# ScenePhotographer: Object-Oriented Photography for Residential Scenes

**Shao-Kui Zhang**
Tsinghua University
Beijing, China
shaokui@tsinghua.edu.cn

**Hanxi Zhu**
Tsinghua University
Beijing, China
13651325353@sina.cn

**Xuebin Chen**
Guangzhou 3D eXtremity
Technology Company
Guangzhou, China
xuebin.chen@3dxt.com

**Jinghuan Chen**
Guangzhou 3D eXtremity
Technology Company
Guangzhou, China
jinghuan.chen@3dxt.com

**Zhike Peng**
Tsinghua University
Beijing, China
pzk22@mails.tsinghua.edu.cn

**Ziyang Chen**
Tsinghua University
Beijing, China
chenziya22@mails.tsinghua.edu.cn

**Yong-Liang Yang**
University of Bath
Bath, United Kingdom
y.yang@cs.bath.ac.uk

**Song-Hai Zhang**[*]
Key Laboratory of Pervasive
Computing, Ministry of Education &
Tsinghua University
Beijing, China
shz@tsinghua.edu.cn

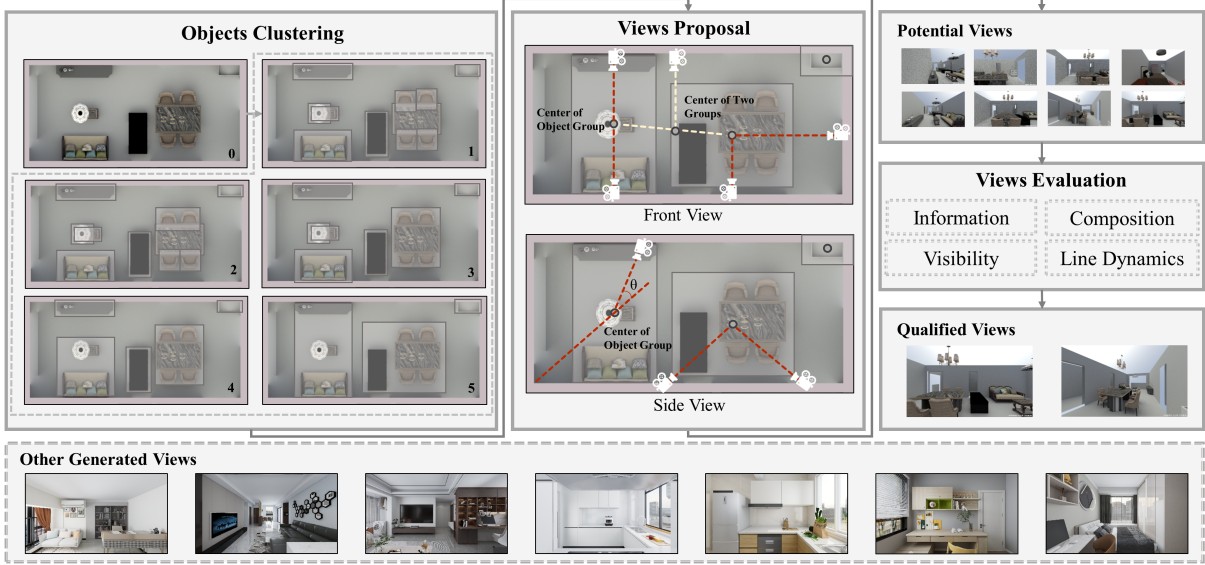

**Figure 1: We present an object-oriented method for automatic view selection in residential scenes. Based on spatial and semantic proximity, we first partition the furniture objects in the room into several groups suitable for observation in the same view (Top-Left). After clustering, we probe one or multiple groups' front and side views to get the potential view set (Top-Middle). Ultimately, we formulate four criteria (Top-Right) to evaluate and select the optimal views (Bottom).**

[*]Corresponding Author.

*MM '24, October 28-November 1, 2024, Melbourne, VIC, Australia*

## Abstract

Humans understand digital 3D scenes by observing them from reasonably placed virtual cameras. Selecting camera views is fundamental for 3D scene applications but is typically manual. Existing

literature on selecting views is based on regular or polygonal room shapes without focusing on the objects in the scene, resulting in poorly composed views concerning objects. This paper introduces ScenePhotographer, an object-oriented framework for automatic view selection in residential scenes. Potential object-oriented views are yielded by a learning-based method, which clusters objects into groups according to objects' functional and spatial relationships. We propose four criteria to evaluate the views and recommend the best batch, including room information, visibility, composition balance, and line dynamics. Each criterion measures the view according to its corresponding photography rule. Experiments on various room types and layouts demonstrate that our method can generate views focusing on coherent objects while preserving aesthetics, leading to more visually pleasing results[1].

## CCS Concepts

• **Computing methodologies** → **Computer graphics**.

## Keywords

View Selection, 3D Interior Scenes, Residential Photography

**ACM Reference Format:**
Shao-Kui Zhang, Hanxi Zhu, Xuebin Chen, Jinghuan Chen, Zhike Peng, Ziyang Chen, Yong-Liang Yang, and Song-Hai Zhang. 2024. ScenePhotographer: Object-Oriented Photography for Residential Scenes. In *Proceedings of the 32nd ACM International Conference on Multimedia (MM '24), October 28-November 1, 2024, Melbourne, VIC, Australia.* ACM, New York, NY, USA, 9 pages. https://doi.org/10.1145/3664647.3680942

## 1 Introduction

Research topics related to virtual 3D scenes have significantly progressed in the last decades, such as scene synthesis [13, 38, 39, 43], scene reconstruction [10, 24, 29], etc. A virtual camera view comprising camera position, orientation, field of view (FoV), and aspect ratio must be carefully selected to better render scenes for human observation and perception.

For instance, in terms of interior 3D scene synthesis, selecting an informative view is one of the most fundamental ways to understand scenes. Typically, researchers fix the view in a predefined top-down direction to showcase and assess the layout results [16, 37, 40]. However, top-down views may not be intuitive for scene understanding, especially the objects therein. As such, interior decoration companies often show their products through eye-level photos. Therefore, automatic view selection contributes to intuitive scene perception for humans.

In addition, rendered scene images can serve as the training data for computer vision tasks, such as semantic segmentation [45]. The selection of views affects the performance of the machine learning model trained on the view-based dataset. Some methods generate training data by selecting views [17, 19], demonstrating that good views contribute to high-quality datasets for deep learning. For furniture retailers, selecting informative and aesthetic views for furniture may leave a lasting impression on potential customers. However, manually taking photos of each product is time-consuming, and it can be difficult for non-experts to specify a desired view. This highlights the commercial significance of automatic view selection.

Nevertheless, only limited approaches to automatic view selection have been proposed. Existing literature primarily focuses on opting views of 3D models [4, 6, 12], with few considerations on object relationships. Moreover, generating views for a 3D model depends on the shape's regularity (e.g., continuity, symmetry), which is unavailable for complex scenes. For scene-level view generation, Zhang et al. [44] introduced a framework that automatically selects informative and aesthetic views for 3D scenes based on interior photography rules. It provides reasonable views based on the shapes of the walls. However, this work needs to consider objects in the scene, such as their positions, types, and interrelations. Thus, it is limited to an intuitive understanding of objects in the scene.

This paper proposes an object-oriented method for automatic view generation in interior scenes. As shown in Fig. 1, our method proposes views for individual object groups in the scene where the objects in the same group are functionally relevant. We consider functional relevance between objects (e.g., a bed and a wardrobe), ensuring that a set of semantically related objects are displayed in the appropriate positions under the same view. As a result, the views generated by our method better exhibit objects and their relations while preserving scene aesthetics.

To generate an object-oriented view, we first group the furniture objects suitable for concurrent observation using an agglomerative clustering algorithm (Section 3). Aiming to divide various objects in indoor scenes functionally, we consider the spatial distance between objects and their quantified "category" differences as the basis for classification. Subsequently, our method generates potential views based on the clustered object groups (Section 4). By applying residential photography rules to the grouping results, such as One-Point Perspective (OPP), Two-Point Perspective (TPP), and Golden Section [20], we can generate a series of appealing views.

However, not all of them are aesthetic and practical. Hence, we further assess and filter the generated views by four criteria from two aspects. First, views are measured by how many object parts are captured (Section 5.1) and how broad the horizon is (Section 5.2) to reveal more scene information. Second, views are measured by how balanced objects are distributed on the canvas (Section 5.3) and how much tension is created by visible objects (Section 5.4) to preserve scene aesthetics.

Extensive experiments on various room types and layouts were conducted to evaluate our method. We compare the views generated by our method, those generated by SceneViewer [44], and those manually created by professional photographers by gathering feedback from human participants (Section 6.2). We also conduct an ablation study to assess the validity of the four criteria (Section 6.3). Moreover, our dataset is collected from an interior design company with thousands of exclusive stores, where our method has been successfully used in real applications for proposing views.

Our technical contributions are as follows:

- We present an object-oriented method for automatically recommending views in residential scenes.
- We propose an algorithm for grouping functional objects based on their spatial distance and semantic category.
- We formulate a set of evaluation criteria to assess the informativeness and aesthetics of views.

---

[1]Please refer to a supplementary video for an overview of our method.

## 2   Related Works

**View Selection for Objects.** View Selection for 3D objects was addressed through several approaches. View entropy was proposed [35] to recommend visual-pleasing and informative views of a single CAD model (e.g., a teapot) or an isolated group of related models (e.g., a desk with a reading lamp and a chair). A benchmark for evaluation of best view selection was introduced [11], but it concentrated on selecting views for a single 3D model presented in triangular meshes. Various methods were proposed for 3D volume [5, 22, 27, 32]. Tao et al. [33] offered a method for selecting views of streamlines. Vázquez proposed a novel method to automatically select an informative view of 3D objects through depth-based view stability analysis [34]. Bruce et al. presented an algorithm based on heuristic compositional rules to find the views with suitable compositions, but the method was toward 3D objects [18]. There is an informative survey [4] for additional studies regarding the view selection for 3D models.

**View Selection for Scenes.** In view selection at the scene level, several methods were proposed for an optimal view of 3D scenes. Early work created a graphical interface to find the desired position of the virtual camera given the frame constraints [3]. A photo assistant was achieved to optimize camera parameters from an initial view to match users' demands on composition [2]. Ji et al. [23] formulated metric functions for the information and aesthetics of views and optimized a set of aesthetically pleasing views through a simulated annealing algorithm. Camera control in cinematography was also explored [7]. Daniyal et al. [8] proposed a method to select the best view of several cameras in the real world by comparing video content scores and human activities.

Furthermore, a recent work applied the rules of residential photography in automatic view selection of the whole scene [44]. However, the rules only concern the shape of walls, which is inconsistent with objects' layouts. In contrast, our method focuses on the functional groups in the room. We distinguish the related objects in the scene and apply rules of residential photography to the groups, automatically recommending aesthetic object-oriented views.

**3D Scenes.** View selection appears in topics related to 3D scenes. For scene synthesis, it is frequent to select a view to visualize the synthesized results [41–43]. For example, user studies need the views to assess the results. For computer vision, view selection generates high-quality images utilizing the views, thus yielding training datasets [17]. Handa et al. [19] generated a training dataset for computer vision through different views from 3D scenes. Luo et al. [25] evaluated 3D scenes using views derived from 3D scenes. Our method automatically recommends views for various scenes, facilitating the related computer graphic and vision studies.

## 3   Object Clustering

In order to find high-quality views at the object level, we first divide indoor objects into groups suitable for observation in the same view. In interior scenes, functionally related furniture objects, such as dining tables and chairs, sofas and coffee tables, etc., may be located close to each other. An agglomerative clustering algorithm is adopted to build a hierarchical organization of objects [1, 28]. However, we alter the definition of "distance" between classes. Suppose $i$ and $j$ represent two objects. $\vec{p_i}$ and $\vec{p_j}$ are their positions.

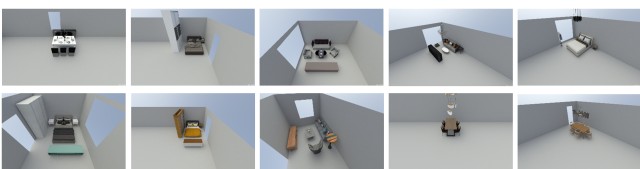

**Figure 2: Manually labelled object groups. Each item contains a group of functionally related objects, such as dining tables and chairs (top left corner).**

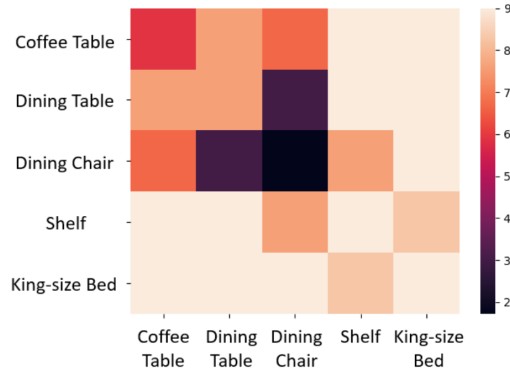

**Figure 3: The heatmap of the "distance of type" between a few classes. The darker the colour is, the closer the distance between the two categories.**

$t_i$ and $t_j$ are their types/categories. The distance between object **i** and object **j** can be formulated as in Equation 1, where $\| \vec{p_i} - \vec{p_j} \|$ is the spatial distance, and $\gamma(\cdot)$ is the "distance of type".

$$D(i, j) = \| \vec{p_i} - \vec{p_j} \| + \gamma(t_i, t_j) \tag{1}$$

Distance of type is based on the probability of two object types appearing in the same functional group. The greater the likelihood that they are considered to be in the same group, the closer they are to the clustering. For example, a dining table and a dining chair are likelier to be in the same group, so their $\gamma(t_i, t_j)$ is lower than that between the dining table and a bunk bed.

We utilize the 3D-Front dataset [14, 15] to obtain the co-occurrence between categories. First, we annotate the objects in the dataset and identify the objects that can be grouped. This allows us to create a dataset of grouped objects, as shown in Fig. 2, where each item contains a group of objects serving the same function.

Using the data above, we calculate the co-occurrence of categories in the same group. $\mathcal{T}(t_i, t_j)$ represents the co-occurrence of $t_i$ and $t_j$. We formulate the distance of type as Equation 2. If an object appears once, we ignore the occurrence with itself.

$$\gamma(t_i, t_j) = -\ln \mathcal{T}(t_i, t_j) \tag{2}$$

We use the negative logarithm to correlate the distance with the co-occurrence frequency and ensure that it has a similar magnitude to the spatial distance. The "distance of type" heatmap is shown in Fig. 3. A darker colour indicates a closer relationship between the two categories. Some grouping results are shown in Fig. 4.

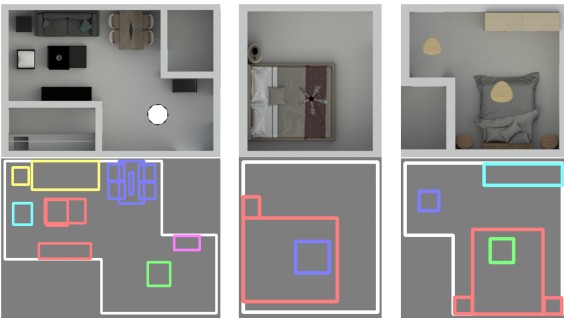

Figure 4: Some results of agglomerative clustering algorithm. The top images show the top-down view of the room, and the bottom images are the corresponding results of their object grouping. The objects in the same group are in the same color, while the white polygons show the shapes of walls.

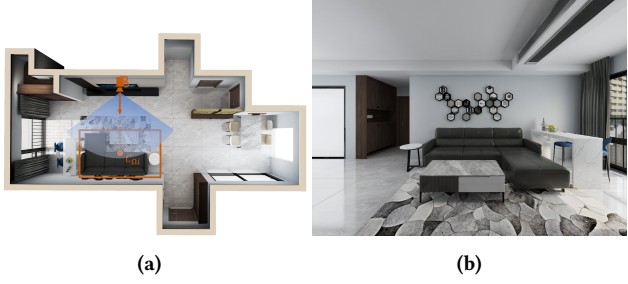

(a)                    (b)

Figure 5: An example of a front view. (a) the camera is positioned in front of the object group and moved back to a wall or an object. (b) the corresponding photograph.

## 4 View Proposal

Subsequently, we can propose views concerning groups of objects. The parameters of a view are determined by $\mathbf{c} = (\vec{\zeta}, \vec{\beta}, \theta, r)$. $\vec{\zeta}$ is the camera position, which is a three-dimensional vector. $\vec{\beta}$ is the camera direction, a three-dimensional normalized vector, i.e., $\|\vec{\beta}\| = 1$. $\theta$ and $r$ are the camera horizontal field of view (HFoV) and aspect ratio, respectively. $r = W/H$ is the ratio of image plane width $W$ and height $H$. For simple notations, we omit the subscripts indexing to individual views.

We explore object groups' front and side views to achieve the OPP and TPP. The front view ensures that most objects are perpendicular or parallel to $\vec{\beta}$, so the rendered image has a single visual vanishing point [20]. $\vec{\rho_i}$ represents the center of the object group $\Omega_i$'s bounding box. We generate four candidate views for each $\Omega_i$, i.e., front, back, left and right. The four directions determine $\vec{\beta}$. $\vec{\zeta}$ starts at $\vec{\rho_i}$ and moves back along $-\vec{\beta}$. The camera goes back until it touches an object or a wall. Fig. 5 shows a front view.

The side view makes objects and the camera form a certain angle, e.g., $45°$, so the rendered image has two vanishing points[20]. In the wall-based view selection, the best position of a camera is located at one of the trisection points of a wall towards the opposite wall[20]. One motivation is that the front wall should occupy a larger area in the canvas than the side wall[44]. In Fig. 6, $\vec{\beta}$ is parallel to the

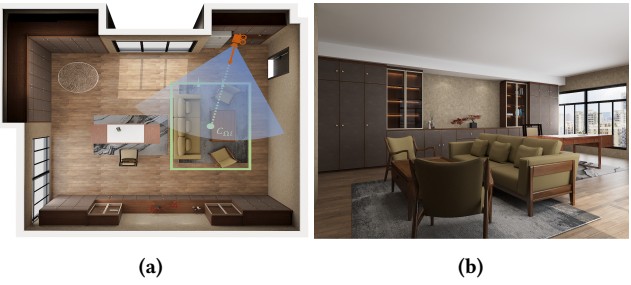

(a)                    (b)

Figure 6: An example of a side view. (a) the camera faces the object group at a slant. (b) the corresponding photograph.

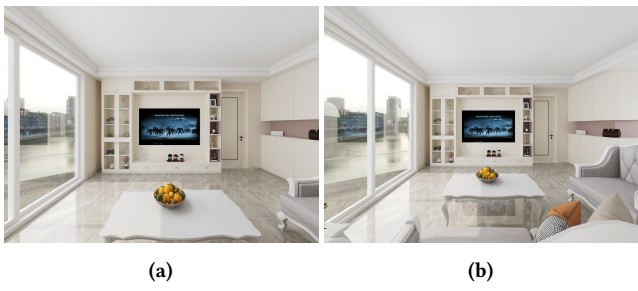

(a)                    (b)

Figure 7: Comparison of different HFoV. (a) a small FoV suits a distant subject. (b) a large FoV suits a nearby subject.

line determined by $\vec{\rho_i}$ and one of the trisection points on the edge of $\Omega_i$'s bounding box. $\vec{\zeta}$ is calculated similarly to the front view.

FoV decides the observation scope of a view. FoV is generally determined by the distance between $\vec{\zeta}$ and the target group. When the target is close to $\vec{\zeta}$, $\theta$ must be increased to perceive things better. When the distance between $\vec{\zeta}$ and the concerned group is far, $\theta$ has to be decreased to avoid viewing other irrelevant objects. Thus, $\theta$ is defined by Equation 3, where function $s(\Omega_i, \vec{\beta})$ returns the projected width of $\Omega_i$ in direction $\vec{\beta}$. $\delta_a$ is an empirical angle to keep $\Omega_i$ a proper ratio in the canvas. Fig. 7 shows an intuitive comparison of a small $\theta$ and a large $\theta$.

$$\theta = 2 \arctan\left(\frac{s(\Omega_i, \vec{\beta})}{\|\vec{\zeta} - \vec{\rho_i}\|}\right) + \delta_a \tag{3}$$

The aspect ratio $r$ is usually a design decision in interior photography. It determines the proportion of the ceiling and floor in the image, generally chosen between $r_1 = 4 : 3$ and $r_2 = 16 : 9$. If a photographer wants to focus on furniture, $16 : 9$ is picked. $4 : 3$ allows more ceiling and floor when other parameters are fixed. Fig. 8 shows the difference. Note that other aspect ratios can be used for the trade-off of including more ceilings/grounds [26], but this is beyond the scope of this paper.

According to the advice of professional photographers, one way to determine $r$ is by calculating the width-to-height ratio of the wall in front of the camera and choosing $r_1$ or $r_2$ based on closeness. $\Lambda_k^{\vec{\zeta},\vec{\beta},\theta}$ represents the visible span of wall $\Lambda_k$ given three parameters. $H$ represents the room's height, and $n_k$ represents the normal vector of $\Lambda_k$. Based on the above advice, $r$ is defined in Equation 4, where

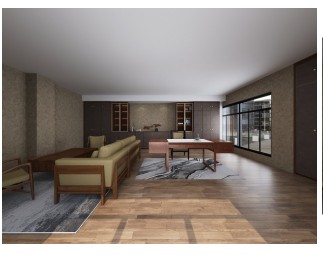

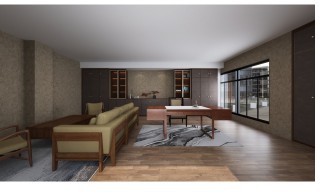

**(a)** $4:3$          **(b)** $16:9$

**Figure 8: Comparison of different aspect ratios. (a) a** $4:3$ **ratio tends to see more ceiling and floor. (b) a** $16:9$ **ratio focuses on walls and objects.**

$\hat{W}$ represents the span of walls in sight measured by Equation 5. $|\overrightarrow{n_k} \cdot \vec{\beta}|$ is walls' contributions concerning the view, e.g., a wall's normal being perpendicular to $\vec{\beta}$ does not contribute to $\hat{W}$.

$$r = \begin{cases} 16:9, \text{ if } \frac{\hat{W}}{H} <= \sqrt{r_1 r_2} \\ 4:3, \text{ otherwise} \end{cases} \tag{4}$$

$$\hat{W} = \sum_k \Lambda_k^{\vec{\zeta},\vec{\beta},\theta} |\overrightarrow{n_k} \cdot \vec{\beta}| \tag{5}$$

# 5 View Evaluation

## 5.1 Room Information

Room information $S_o$ measures how many objects can be seen and how complete each object is. Existing works [35, 36, 44] count the number of object primitives in a view (i.e., check whether the view captures an object primitive). However, the geometric complexity of objects is different. A large object can be geometrically simple with few primitives, but it occupies a prominent part of the view. Thus, we measure the room information incorporating object volume.

Similar to HAISOR [31], each object $o_i \in O$ in our method is recognized as several parts $a_{i,j} \in o_i$, e.g., handrails, sliders etc. $O = \{o_1, o_2, ..., o_{N_o}\}$ contains the objects in the room. Calculating detailed part geometry is too heavy, so we use a bounding box to represent each part. Precisely, the room information $S_o$ is measured by Equation 6, taking as input a view $\mathbf{c}$ and the objects in room $O$.

$$S_o(\mathbf{c}, O) = \sum_{i,j} \frac{f_o(\mathbf{c}, a_{i,j}) V(a_{i,j})}{\|\mathcal{P}(a_{i,j}) - \vec{\zeta}\|} \tag{6}$$

$f_o(\mathbf{c}, a_{i,j})$ returns the number of bounding box's vertices in $a_{i,j}$ seen by view $\mathbf{c}$. $f_o$ tests each point and counts the number of points passing the tests. It first tests if the point is projected on the camera frustum's horizontal and vertical visual planes, which are determined by the FoV $\theta$ and aspect ratio $r$. Then, it tests whether occlusions exist between the camera position $\vec{\zeta}$ and the part $a_{i,j}$. $V(a_{i,j})$ calculates the volume of the given part $a_{i,j}$, i.e., how large an object part is. Therefore, if an object is sufficiently large and mostly visible, the object will largely contribute to the room information. The denominator in Equation 6 is the distance between the view position $\vec{\zeta}$ and the part position $\mathcal{P}(a_{i,j})$, indicating that objects closer to the camera appear larger and vice versa. Generally,

if a view captures sufficient objects and the objects are sufficiently large on the image plane, the view's room information will be high.

Fig. 9a illustrates the intuition. With more volumes of large objects captured, an informative view can be selected.

## 5.2 Visibility

According to [21], a view should be set as far as possible, introducing a sense of room "depth". In the rendering process, the view casts rays to its image plane, where the "depth" is determined by each ray's first hit of the scene. Ideally, most rays would hit walls and objects, while some may travel a sufficiently long distance until a far corridor, window, or door is hit. The sense of room depth needs to be larger under this criterion.

The visibility $S_v$ is measured by Equation 7. It is formulated as an integration over the image plane, where $w$ and $h$ define a ray originates from the camera position $\vec{\zeta}$ and passes through a position $(w, h)$ on the image plane. $f_v(\cdot)$ is the ray-casting operation in computer graphics. Therefore, $f_v(\cdot)$ returns the closest ray-scene intersection point, and $\|\cdot\|$ is the distance. q is the hyper-parameter greater than 1. It penalizes the casted positions too close to the camera while rewarding the rest.

Fig. 9b shows the intuition of visibility. A visible view can be selected with more rays penetrating deeper into a room.

$$S_v(\vec{\zeta}, O \cup \mathcal{R}) = \int_{w,h} (\|\vec{\zeta} - f_v(\vec{\zeta}, w, h, O \cup \mathcal{R})\|)^q \mathrm{d}w \mathrm{d}h \tag{7}$$

## 5.3 Composition Balance

Composition balance measures how objects deviate from the view center. According to the law of symmetry in gestalt psychology, a view with objects evenly distributed on both sides enhances the coordination of the user's perception [30]. An extreme case with total imbalance is when all objects are captured on the left side of the image. Since most objects in residential scenes are near the ground, we mainly measure the horizontal deviations [26].

The composition balance $S_b$ is measured by Equation 8, where objects are not split into parts. It is a weighted sum concerning object volume $V(o_i)$, i.e., the imbalance of a large object affects $S_b$ more than that of a small object. $f_b(\mathbf{c}, o_i)$ measures the signed distance from an object to the image center as shown in Equation 9, where $\mathcal{I}(o_i)$ returns the projected positions of object $o_i$ on the image, $x$ takes the horizontal entry, and $W$ is the image width. $f_b(\mathbf{c}, o_i)$ is further scaled by the distance between the view position $\vec{\zeta}$ and the object position $\mathcal{P}(o_i)$, indicating that distant objects have less impact on $S_b$. The final exponential operation $exp(-(\cdot))$ ensures that $S_b$ decreases as objects become imbalanced since a perfectly balanced image yields 0 of a weighted sum over $f_b(\mathbf{c}, o_i)$.

$$S_b(\mathbf{c}, O) = exp(-\frac{\sum_i (V(o_i) \frac{f_b(\mathbf{c}, o_i)}{\|\mathcal{P}(o_i) - \vec{\zeta}\|})}{\sum_i V(o_i)}) \tag{8}$$

$$f_b(\mathbf{c}, o_i) = \frac{W/2 - \mathcal{I}(o_i)^x}{W/2} \tag{9}$$

Fig. 9c shows the intuition of composition balance. A view is balanced if objects are equally distributed between the vertical visual plane concerning their volume.

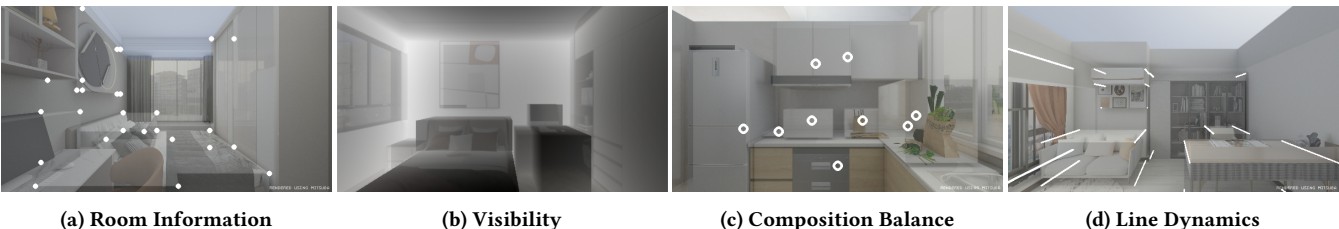

(a) Room Information      (b) Visibility      (c) Composition Balance      (d) Line Dynamics

**Figure 9: Illustrating how existing content contributes to the criteria. (a) Room information captures more objects, object parts and their volumes. (b) Visibility preserves more space in front of the view. (c) Composition balance prefers distributing objects uniformly on the image plane concerning objects' volumes. (d) Line dynamics measures the tension formed by objects' lines.**

## 5.4 Line dynamics

Line dynamics is one of the most significant mechanisms to create a sense of space in views [9]. The impact of a view is enhanced when its borders are broken at oblique angles by sufficient lines, which refer to objects' outer contours in Fig. 9d. Due to perspectives, the lines are parallel in 3D but are gathered at one vanishing point in 2D. Thus, we count the lines associated with the vanishing point to measure the line dynamics. An image is more expressive if the angles between the lines and the visual plane are close to $\pi/4$ [9].

Assume the objects' bounding boxes in a room derive a line set $\mathcal{L}(\mathbf{c}, O) = \{l_1, l_2, ..., l_{N_l}\}$. Note that the lines are already projected on the image. Each line $l_k$ passes through a point $(l_k^x, l_k^y)$ with angle $\tau_k$, the line dynamics are calculated by Equation 10. It counts the number of lines intersecting the vanishing points, while each line is assigned a weight indicating its contribution to the expression.

$$S_l(\mathcal{L}(\mathbf{c}, O)) = \sum_{k=1,2,...,N_l} \alpha(\tau_k) \times \begin{cases} 1, & \text{if } \mathbf{d}(l_k, \mathbf{v}) \le \delta_l \\ 0, & \text{otherwise} \end{cases} \quad (10)$$

$$\alpha(\tau_k) = \cos(2 \times \min\{|\tau_k + n * \pi/2 - \pi/4|, n = 0, 1, 2, 3\}) \quad (11)$$

$$\mathbf{d}(l_k, \mathbf{v}) = |\cos(\tau_k)(l_k^x - \mathbf{v}^x) - \sin(\tau_k)(l_k^y - \mathbf{v}^y)| \quad (12)$$

$\alpha(\tau_k)$ addresses the expression in Equation 11. It first calculates the minimal angle between line $l_k$ and the four $\pi/4$ lines. If line $l_k$ aligns perfectly with a $\pi/4$ line, then $\cos(2(\cdot)) = 1$ indicating a total contribution to $S_l$. If line $l_k$ is "upright", i.e., being perpendicular to the ground, then $\cos(2(\cdot)) = 0$ even if $l_k$ is sufficiently close to the *main* vanishing point as shown in Equation 12. Since a view may have more than one vanishing point, we define $\mathbf{v} = (\mathbf{v}^x, \mathbf{v}^y)$ as the *main* vanishing point with most lines intersecting. As lines may have multiple vanishing points (i.e., lines are not parallel), the view's vanishing points are the means of each line's vanishing points. Equation 12, therefore, calculates the distance from line $l_k$ to $\mathbf{v}$. If line $l_k$ is sufficiently close to $\mathbf{v}$ concerning a threshold $\delta$, line $l_k$ contributes to $S_l$. Otherwise, line $l_k$ does not contribute even though Equation 11 returns a non-zero value.

Fig. 9d shows the intuition of line dynamics. A deeply satisfying, resolved feeling is produced with more objects contributing their lines perpendicular to the image plane.

## 6 Experiments

## 6.1 Setup and Results

The dataset is provided by Guangzhou 3D eXtremity Technology Company with over 1000 dealers and 2500 exclusive shops for residential home customization. The dataset includes floor plans, furniture objects, and their arrangements created by professional interior designers for commercial purposes. The threshold in our agglomerative clustering algorithm is set to $S/3 + 4.5$, where $S$ represents the larger value between the room width and length.

$\delta_a$ in Equation 3 is $10°$. $\delta_l$ in Equation 10 is $1.0 \times 10^{-3}$. The weight of room information is 1.0. The weight of visibility is 0.5. The weight of composition balance is 10.0. The weight of line dynamics is 2.0. The criteria weights primarily uniformize various scales. All criteria are effective with a balanced scale. The parameters are not data sensitive because indoor scenes and furniture objects generally have the same scale of spatial dimensions.

In order to improve the efficiency of the visibility test, we simplify the integration by summing up evenly spaced samples (15-pixel intervals in our implementation) on the canvas.

As shown in Fig. 10, our method generates aesthetically pleasing views focusing on coherent objects and can be applied to various room types. Our method is successfully integrated into the actual application of the interior design company. When users search for rooms or floorplans, the thumbnails are rendered according to our views. The interior designers also use our method to automatically generate views after designing a new scene. Our method also applies to 3D-Front [14], one of the latest and most fundamental datasets for scene synthesis [31]. Please see the supplementary materials for the results generated from 3D-Front. A 3D-Front room with 6-10 objects consumes less than 10 seconds to generate views, which is much more efficient than manual view selection.

## 6.2 User Study

We conducted a user study to evaluate the effectiveness of our method by comparing it with views generated by professional photographers and SceneViewer [44]. We invited 6 photographers with solid interior design and photography backgrounds to generate images manually. They were asked to adjust the visual camera to find the best views interactively. Overall, we obtain 92 such images.

To evaluate the results, we invited two groups of participants to compare views generated by our method, photographers and SceneViewer. The first group, referred to as the "professional group", consists of professional photographers with backgrounds similar to those who created the manual views. The second group, referred to as the "third-party group", consists of university students knowledgeable about arts, photography, or interior design. Each newly recruited participant was presented with a questionnaire. Each

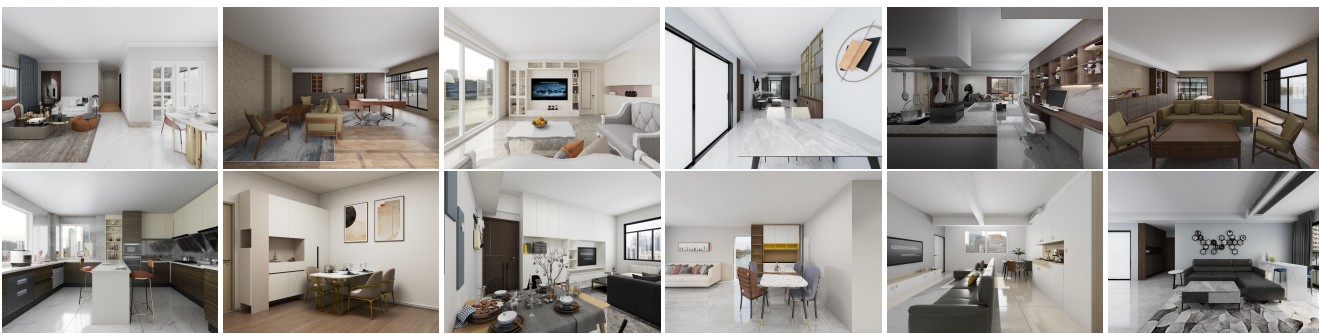

**Living Rooms, Dining Rooms and Kitchens.**

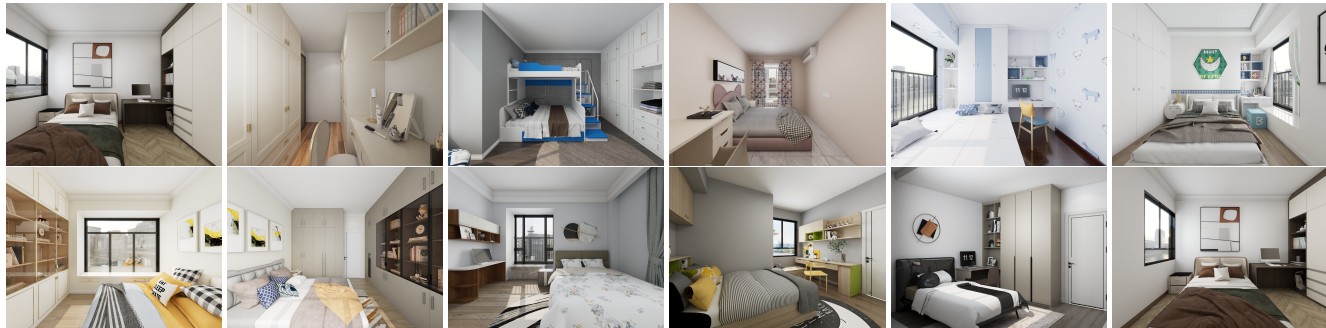

**Bedrooms, Dressing Rooms and Kids Rooms.**

**Figure 10: Views generated by our method. More results are shown in the supplementary materials.**

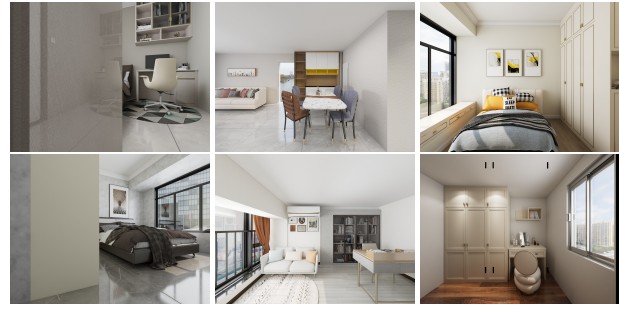

(a) SceneViewer [44]          (b) Our Method          (c) Photographers

**Figure 11: Qualitative results. (a) SceneViewer [44] is reliant on walls. Views can only be derived from the room shape. (b) Our method generates views concerning object groups (Top) and aesthetic constraints (Bottom for Line Dynamics). (c) Professional Human Photographers may manually remove a few unnecessary walls or items to augment views.**

question displayed three images generated in different ways for a specific room, where the images were shown in random order.

Participants compared three images and rated them based on aesthetics, informativeness, and overall quality. The aesthetics refers to how participants feel a sense of beauty in images, with 0 being "terrible taste" and 5 being "fine art". The informativeness refers to how participants receive sufficient details in images, with 0 being "no idea" and 5 being "rich details". The overall quality asks for a comprehensive rating, with 0 being the worst and 5 being the best.

For each questionnaire, rooms were randomly assigned, and the order of the rooms was also random. For the professional group, 5 photographers were invited. Each photographer was assigned more than 500 questions (taking approximately 8 hours to finish), and we achieved 2933 scores for each image. For the third-party group, 31 knowledgeable participants were invited. Each one was assigned 45 questions, and we achieved 1395 scores.

The comparison results are shown in Table 1 for the professional group and Table 2 for the third-party group. We noticed that the professional group is stricter than the third-party group in evaluating the views. However, both groups indicate that, though the photos taken by professional human photographers are better than ours, our method performs better than the prior art on 3D scene photography [44] and is competitive with photographers. Several Kruskal-Wallis Tests show that the differences between methods (e.g., Ours versus SceneViewer) are statistically significant, with the p-values being extremely close to 0.

Fig. 11 shows a few qualitative comparisons. SceneViewer [44] largely depends on walls to yield views. Thus, it is hard for SceneViewer to address irregular shapes. Our object-oriented method helps locate plausible camera positions to avoid obstruction of walls. It directly focuses on object groups under photography constraints, e.g., selecting views with good object composition. Professional photographers may remove a wall or an object to acquire more space for better view specification.

## 6.3 Ablation Study

We conducted an ablation study to assess the validity of the four criteria. After completing the previous experiment, the participants

**Table 1: Professional evaluation. The participants are professional interior photographers. Each cell contains an average score and a standard deviation. Each row refers to a baseline. "Human" refers to professional photographers in Section 6.2.**

| Methods | Aesthetics | Informativeness | Overall |
| --- | --- | --- | --- |
| SceneViewer | 1.62 (0.987) | 1.75 (0.716) | 1.54 (0.802) |
| Ours | 2.35 (0.667) | 2.58 (0.659) | 2.46 (0.666) |
| Human | 2.78 (0.959) | 3.08 (0.783) | 2.90 (0.95) |

**Table 2: Third-party evaluation similar to Table 1. "Human" stands for knowledgeable participants in Section 6.2.**

| Methods | Aesthetics | Informativeness | Overall |
| --- | --- | --- | --- |
| SceneViewer | 2.60 (1.25) | 2.66 (1.35) | 2.58 (1.09) |
| Ours | 3.67 (0.697) | 3.77 (0.719) | 3.68 (0.593) |
| Human | 4.14 (0.599) | 4.31 (0.608) | 4.17 (0.565) |

**Table 3: Ablation Study. "RoomInfo", "Visibility", "CompBal", and "LineDyna" refer to the criteria in Sections 5.1, 5.2, 5.3 and 5.4, respectively. Each cell contains an average score and a standard deviation. "All" refers to views with all criteria, and other individual rows have a criterion discarded.**

| Methods | Aesthetics | Informativeness | Overall |
| --- | --- | --- | --- |
| No RoomInfo | 3.43 (0.690) | 3.15 (0.679) | 3.27 (0.796) |
| No Visibility | 2.91 (0.509) | 2.83 (0.539) | 2.72 (0.648) |
| No CompBal | 3.27 (0.523) | 3.08 (0.697) | 3.20 (0.717) |
| No LineDyna | 3.10 (0.500) | 2.91 (0.863) | 3.04 (0.796) |
| ALL | 3.84 (0.609) | 3.90 (0.688) | 3.78 (0.587) |

were asked to compare the final results with views without considering one of the criteria. Each participant conducted a questionnaire, where each question had two unordered views from a randomly selected room, including one view taking into account all criteria and the other with one of the criteria ablated. We also asked each participant for ratings regarding each view's aesthetics, informativeness, and overall quality.

The results are shown in Table 3. The views with all criteria outperform those ablating one criterion, indicating that the proposed criteria are all necessary to yield plausible views. Kruskal-Wallis Tests show that the differences between "all criteria assembled" and "discarding a criterion" are statistically significant.

We notice that removing the visibility largely affects the results as visibility requires sufficient space in front of the camera. Without considering visibility, a few objects will be placed out of space, resulting in a nearly empty view. We also find that ablating a specific criterion influences both aesthetics and informativeness. This is understandable because aesthetics and informativeness are often

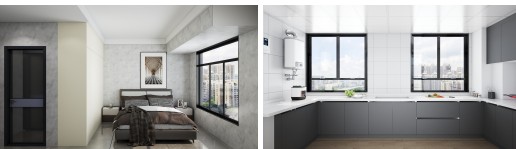

**Figure 12: Photos manually taken by removing walls to create more space. Small rooms or corridors significantly benefit from such operation.**

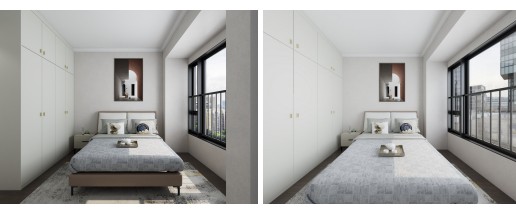

|  |  |
| --- | --- |
| **(a) Not Distorted** | **(b) Distorted** |

**Figure 13: A view (a) is distorted by enlarging its FoV (b), e.g., the wardrobe and bed are "stretched".**

coupled. For instance, an aesthetic view can be attributed to well-selected objects, which is also informative.

## 7 Conclusions

We present ScenePhotographer, a novel method for photographing digital 3D scenes based on object clustering, view proposal, and view selection in an object-oriented manner. Extensive evaluations demonstrate the effectiveness of our method for generating plausible views of 3D scenes. Our codes are publicly available[2]. The view selection criteria can be used as a general function to evaluate scene views quantitatively. We expect our work to further contribute to applications based on 3D scenes and their 2D images.

While achieving quality results, our method has some limitations. First, our work does not explore all possible views as it is computationally prohibitive. Physical entities such as walls and furniture objects may not be easily moved. Hence, the space is often limited when placing the camera. However, virtual scenes can be easily moved or even removed if needed. Some professional photographers took photos with wall removed such that more spaces are available for a better camera position (i.e., $\mathbf{c} = (\vec{\zeta}, \vec{\beta}, \theta, r)$ in Section 4). Fig. 12 shows two photos taken by removing walls, which is particularly helpful for displaying small rooms.

Also, by interviewing the professional photographers in Section 6.2, they suggested that we should prevent the camera from being too close to the objects, i.e., the distance from the camera to a particular object group should be above a threshold. This is because our method enlarges FoV when getting closer to the objects. However, a large FoV potentially causes distortions, i.e., furniture objects may be distorted in the image plane, as shown in Fig. 13. Besides, a view should present a sense of space. Being too close to an object diminishes the surrounding space, thus making it difficult to display the whole scene.

---

[2]https://github.com/Shao-Kui/3DScenePlatform#scenephotographer.

## Acknowledgments

This work was supported by the National Key Research and Development Program of China (No. 2023YFF0905104), the National Natural Science Foundation of China (No. 62132012, 62361146854) and Tsinghua-Tencent Joint Laboratory for Internet Innovation Technology. Yong-Liang Yang is funded by UKRI grant CAMERA (No. EP/T022523/1). Shao-Kui Zhang is funded by Shuimu Tsinghua Scholar Program (No. 2023SM061), China Postdoctoral Science Foundation (No. 2024M751696), Postdoctoral Fellowship Program of CPSF (No. GZB20230353), Tsinghua University Student Research Training (No. 2421T0278, 2421T0277, 2411T0372, 2411T0371) and Young Elite Scientists Sponsorship Program by BAST (No. BYESS2024242).

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
