# OpenReview forum: "ScenePhotographer: Object-Oriented Photography for Residential Scenes"
_acmmm.org/ACMMM/2024/Conference — MM2024 Oral_

### Official Review · Reviewer_7gx4 · 2024-05-17

**Rating:** 4
**Confidence:** 3

**Summary:**

This paper proposes an automatic selection method for camera views that are likely to be appealing in digital residential scenes. The proposed method generates multiple view proposals showing objects and their relationships, and selects the view with the maximum score calculated by four criteria: room information, visibility, composition balance, and line dynamics. Experiments show that the proposed method achieves better scores than the conventional method based on aesthetics, informativeness, and overall quality in user studies.

**Strengths:**

1. The paper is well-written, and the overall idea is easy to follow.
2. The problem addressed is an important and interesting one. While much previous work generates views based on room shapes, this paper focuses on the functional and spatial relationships of objects to generate views.
3. The proposed method achieved the desired results on a dataset provided by an interior design company. Furthermore, the ablation study demonstrated the validity of the proposed four criteria.

**Limitations:**

1. The proposed method requires floor plans, furniture objects, and their arrangements and type labels in digital residential scenes. This rich information is not necessarily available for every scene.
2. Object clustering in the proposed method relies on heat maps of distances of types built in advance, and the number of possible types is 35 according to Figure 3. The objects to which the proposed method can be applied are limited to those present in this heat map.
3. Experiments lack verification of the influence of various parameters of the proposed method, such as room information weights. The authors should explain how the values of these parameters were determined.
4. In Section 1, the authors highlight the importance of automatic view selection by pointing out the time-consuming nature of manual view selection. Nevertheless, the experiment does not evaluate the consumption time of view selection.

**Suitability:**

2

---

### Official Review · Reviewer_MS94 · 2024-05-28

**Rating:** 5
**Confidence:** 2

**Summary:**

This paper introduces a method for automatic view generation in synthetic residential scenes. The method integrates functionality with photographic principles to produce informative and aesthetically pleasing views. The method consists of three core stages: firstly, object clustering is performed to identify groups of objects that can be captured from a single viewpoint and that correlate with each other. Secondly, using photographic rules and the identified object groups, potential views are proposed. Finally, each view is evaluated based on predefined quality metrics, and the best views are selected.

**Strengths:**

Overall, I find this paper to be very well-executed, with good results and a well-explained method.

In the following, I list the strengths of the paper:
- Each component of the method is well justified and thoroughly motivated
- The approach is clearly and precisely explained, with simple equations supported by informative visuals.
- The method is simple. The paper provides enough implementation details (+ code is released).
- A comprehensive user study significantly demonstrates its superiority over state-of-the-art (SceneViewer).
- An ablation study validates the relevance of each „view evaluation metric.“
- The supplementary material contains extensive results, and the video is clear, providing a comprehensive overview.
- While each part in isolation might not be highly novel, their combination presents sufficient novelty, making the approach valuable for the community.
- I see great potential for generating improved synthetic datasets, where information is maximized and redundancy minimized.

**Limitations:**

The paper has only a few limitations that could be addressed in a revised version to enhance its comprehensiveness:

- Direct comparisons between ScenePhotographer, SceneViewer, and human-generated views are missing. Fig. 11 shows comparisons on different scenes, but it would be more insightful to see how ScenePhotographer improves the view within the same scene.
- It would be interesting to provide more insights about the professional camera choice and ScenePhotographer; how much do the parameters deviate from each other on the same scene?
- It would be valuable to see how the „view evaluation metrics“ assess the views created by SceneViewer and professional photographers. Can the proposed metrics serve as a general evaluation function? For instance, do professional views not only receive higher scores in the user study but also achieve higher evaluation scores using these metrics?
- A related question is whether the method is limited by the potential view generation or by the quality of the view evaluation metrics.

minors:
- Fig. 9: An overlay between the image and mask could improve clarity. For example, it was challenging to clearly connect each dot to the corresponding object in Fig. 9(a)
- Fig. 3: Why does the diagonal of the heatmap not show a smaller „distance of type“? Is it because some objects do not appear multiple times in the same scene?
- Fig. 3: It would be more informative to show the heatmap between 4-5 objects instead of the full heatmap without any labels or add the class labels.

**Suitability:**

3

---

### Official Review · Reviewer_v589 · 2024-06-03

**Rating:** 3
**Confidence:** 1

**Summary:**

The paper presents a methods for automatic view selection when taking photograph of a 3d residential scene. It builds on top of a previous work [44] with two additions: (1) objects clustering so that objects with similar functionality will be seen from a similar view, (2) four additional aesthetic criteria for further selection from candidate views.

The work seems to be one reasonably nice step further on top of [44]. My major concern is that the paper does not fit the conference so much, as it is pretty much unimodal.

**Strengths:**

+ The idea of object clustering based on functionality is promising. It further considers the relationship among the objects in the scene from a semantic point of view, which might better match human's intention in certain applications (e.g., interior design).
+ The four proposed criteria seems to work properly, although it is hard to judge whether these are somewhat the optimal group of criteria.

**Limitations:**

As mentioned earlier, the main concern is the mismatch of focus of the paper versus the scope of the conference.

**Suitability:**

1

---

### Meta-Review · Area_Chair_KjG6 · 2024-07-03

**Recommendation:** Accept (Oral)
**Confidence:** 4

**Metareview:**

The paper proposes an automatic camera view selection method for digital residential scenes. It extends a previous work [44] via object clustering and aesthetic criteria. The reviewers agree that the paper is well-written and the proposal is technically sound. Moreover, the authors have adequately responded to the reviewers' concerns in the rebuttal, including an extended discussion of the possible limitations. Although one of the reviewers is concerned with the paper's suitability for the scope of the conference, he/she has recognized that it might be interesting at least for part of the ACMMM audience. Thus, after a thorough evaluation by the reviewers and considering the authors' detailed rebuttal, I recommend accepting this paper as a poster presentation at ACM MM 2024.